

# Complex interplay between stress perturbations and viscoelastic relaxation in a two-asperity fault model

Emanuele Lorenzano and Michele Dragoni

Dipartimento di Fisica e Astronomia, Alma Mater Studiorum Università di Bologna, Viale Carlo Berti Pichat 8, 40127 Bologna, Italy.

*Correspondence to:* Emanuele Lorenzano (emanuele.lorenzano2@unibo.it)

**Abstract.** We consider a plane fault with two asperities embedded in a shear zone, subject to a uniform strain rate owing to tectonic loading. After an earthquake, the static stress field is relaxed by viscoelastic deformation. We treat the fault as a discrete dynamical system with three degrees of freedom: the slip deficits of the asperities and the variation of their difference due to viscoelastic deformation. The dynamics of the system is described in terms of one sticking mode and three slipping modes. We consider the effect of stress transfers connected to earthquakes produced by neighbouring faults. The perturbation is studied in terms of a vector in the state space, whose components are the changes in the state variables of the system. The interplay between the stress perturbation and the viscoelastic relaxation significantly complicates the evolution of the fault and its seismic activity. We show that the presence of viscoelastic relaxation prevents any simple correlation between the change of Coulomb stresses on the asperities and the anticipation or delay of their failures. As an application, we study the effects of the 1999 Hector Mine, California, earthquake on the post-seismic evolution of the fault that generated the 1992 Landers, California, earthquake, which we model as a two-mode event associated with the consecutive failure of two asperities.

## 1 Introduction

Asperity models have long been acknowledged as an effective means to describe many aspects of fault dynamics (Lay et al., 1982; Scholz, 2002). In such models, it is assumed that the bulk of energy release during an earthquake is due to the failure of one or more regions on the fault characterized by a high static friction and a velocity-weakening dynamic friction. The stress build-up on the asperities is governed by the relative motion of tectonic plates. Earthquakes that have been ascribed to the slip of two asperities are the 1964 Alaska earthquake (Christensen and Beck, 1994), the 1992 Landers, California, earthquake (Kanamori et al., 1992), the 2004 Parkfield, California, earthquake (Twardzik et al., 2012) and the 2010 Maule, Chile, earthquake (Delouis et al., 2010).





In the framework of asperity models, a critical role is played by stress accumulation on the asperities, fault slip at the asperities and stress transfer between the asperities. Accordingly, fault dynamics can be fruitfully investigated via discrete dynamical systems whose essential components are the asperities (Ruff, 1992; Turcotte, 1997). Such an approach reduces the number of degrees of freedom required to describe the dynamics of the system and allows the study of its evolution by means of orbits in the phase space and a finite number of dynamic modes. Asperity models are capable to reproduce the essential features of the seismic source, while sparing the more complicated characterization based on continuum mechanics.

In a number of recent works, modelling of different mechanical phenomena in a two-asperity fault system has been addressed, such as stress perturbations due to surrounding faults (Dragoni and Piombo, 2015) and the radiation of seismic waves (Dragoni and Santini, 2015). In these models, the fault is treated as a discrete dynamical system with four dynamic modes: a sticking mode, corresponding to stationary asperities, and three slipping modes, associated with the separate or simultaneous failure of the asperities.

The impact of viscoelastic relaxation has first been studied by Amendola and Dragoni (2013) and then further investigated by Dragoni and Lorenzano (2015), who considered a fault with two asperities of different strengths. The authors discussed the features of the seismic events predicted by the model and showed how the shape of the associated source functions is related to the sequence of dynamic modes involved. In turn, the observation of the moment rate provides an insight on the state of the system at the beginning of the event.

However, no fault can be considered isolated; in fact, any fault is subject to stress perturbations associated with earthquakes on neighbouring faults (Harris, 1998; Stein, 1999; Steacy et al., 2005). Whenever a fault slips, the stress field in the surrounding medium is altered. As a result, the occurrence time and the magnitude of next earthquakes may change with respect to the unperturbed condition, which is governed by tectonic loading.

The aim of the present paper is to discuss the combined effects of viscoelastic relaxation and stress perturbations on a two-asperity fault in the framework of a discrete fault model. In order to deal with such a problem, we base on the results achieved by Dragoni and Piombo (2015) and Dragoni and Lorenzano (2015). In the former work, the authors considered a two-asperity fault with purely elastic rheology and discussed the effect of stress perturbations due to earthquakes on neighbouring faults. The fault was treated as a discrete dynamical system whose state is described by two variables, the slip deficits of the asperities. In the latter work, viscoelastic relaxation on the fault was dealt with by adding a third state variable, the variation in the difference between the slip deficits of the asperities during interseismic intervals. In the present paper, we introduce stress perturbations as modelled by Dragoni and Piombo (2015) in the framework of the two-asperity fault considered by Dragoni and Lorenzano (2015). Accordingly, the present work represents a three-dimensional generalization of the model devised by Dragoni and Piombo (2015). As further complications with respect to previous works, elastic wave radiation and additional constraints on the state of the fault are taken into account.



In the framework of present model, seismic events generated by the fault are discriminated according to the number and sequence of slipping modes involved and the seismic moment released; these features are related to the particular state of the system at the beginning of the interseismic interval preceding the event. We discuss how stress perturbations affect the evolution of the fault in terms of changes in the state of the system and in the duration of the interseismic time, highlighting

the complications arising from the ongoing post-seismic deformation process with respect to the purely elastic case considered by Dragoni and Piombo (2015). As an application, we consider the stress perturbation imposed by the 1999 Hector Mine, California, earthquake (Jónsson et al., 2002; Salichon et al., 2004) to the fault that originated the 1992 Landers, California, earthquake, which we model as a two-mode event due to the consecutive failure of two asperities and that was followed by remarkable viscoelastic relaxation (Kanamori et al., 1992; Freed and Lin, 2001). We propose a means to estimate the stress

transfer from the knowledge of the relative positions and faulting styles of the two faults. As a further novelty with respect to the work presented by Dragoni and Lorenzano (2015), we show how the knowledge of the time interval elapsed after the 1999 earthquake can be used to constrain the admissible set of states that may have given rise to the 1992 event. We discuss the possible subsequent evolution of the Landers fault after the stress transfer from the Hector Mine earthquake, pointing out the main differences with respect to an unperturbed scenario.

**2    The model**

We consider a plane fault containing two asperities of equal areas $A$ and different strengths that we name asperity 1 and asperity 2 (Fig. 1). The fault is enclosed in a homogeneous and isotropic shear zone behaving as a Poisson solid and is subject to a uniform strain rate owing to the relative motion of two tectonic plates, taking place at a constant rate $V$. As for the rheology of lithospheric rocks, we assume a Maxwell viscoelastic behaviour with a characteristic relaxation time $\Theta$. Finally, we character-

ize the seismic efficiency of the fault by means of an impedance $\gamma$. All quantities are expressed in nondimensional form.

In accordance with the assumptions of asperity models, we ascribe the generation of earthquakes on the fault to the failure of the sole asperities, neglecting any contribution of the surrounding weaker region to the seismic moment. Also, we do not describe friction, slip and stress at every point of the fault, but only consider their average values on each asperity.

The fault is treated as a dynamical system with three state variables, functions of time $T$: the slip deficits $X(T)$ and $Y(T)$ of asperity 1 and 2, respectively, and the variable $Z(T)$ representing the variation of the difference between the slip deficits of the asperities, owing to the viscoelastic rheology of lithospheric rocks. The slip deficit of an asperity is defined as the slip that an asperity should undergo at a given instant in time in order to recover the relative displacement of tectonic plates occurred

up to that moment.

We assume the simplest form of rate-dependent friction and associate the asperities with constant static and dynamic frictions, the latter considered as the average value during slip. The static friction on asperity 2 is a fraction $\beta$ of that on asperity



and dynamic frictions are a fraction $\epsilon$ of static frictions for both asperities. Letting $f_{s1}$ and $f_{d1}$ be the static and dynamic frictions on asperity 1, respectively, and $f_{s2}$ and $f_{d2}$ be the static and dynamic frictions on asperity 2, respectively, we have

$$\beta = \frac{f_{s2}}{f_{s1}} = \frac{f_{d2}}{f_{d1}}, \qquad \epsilon = \frac{f_{d1}}{f_{s1}} = \frac{f_{d2}}{f_{s2}}. \tag{1}$$

In units of static friction on asperity 1, the tangential forces on the asperities during a global stick mode are

$$F_1 = -X + \alpha Z, \qquad F_2 = -Y - \alpha Z. \tag{2}$$

In these expressions, the terms $-X$ and $-Y$ represent the effect of tectonic loading and have the same sign for both asperities, whereas the terms $\pm\alpha Z$ are the contributions of stress transfer between the asperities, in the presence of viscoelastic relaxation. The forces are applied in the slip direction. The parameter $\alpha$ is a measure of the degree of coupling between the asperities.

An effective way to characterize fault mechanics is provided by the concept of Coulomb stress (Stein, 1999). It is defined as the difference between the shear stress $\sigma_t$ in the direction of fault slip and the static friction $\tau_s$ on the fault surface:

$$\sigma_C = \sigma_t - \tau_s. \tag{3}$$

Accordingly, $\sigma_C$ is negative during an interseismic interval and a seismic event occurs when $\sigma_C$ vanishes. In our model, the presence of two asperities makes it necessary to assign a value of Coulomb stress to each of them. By definition, the Coulomb

forces on asperity 1 and 2 are, respectively,

$$F_1^C = -F_1 - 1, \qquad F_2^C = -F_2 - \beta. \tag{4}$$

Using Eq. (2), they can be rewritten as

$$F_1^C = X - \alpha Z - 1, \qquad F_2^C = Y + \alpha Z - \beta. \tag{5}$$

To sum up, the system is described by the set of six parameters $\alpha, \beta, \gamma, \epsilon, \Theta$ and $V$, with $\alpha \geq 0$, $0 < \beta < 1$, $\gamma \geq 0$, $0 < \epsilon < 1$,

$\Theta > 0$ and $V > 0$. At any instant $T$ in time, the state of the system may be univocally expressed by the tern $(X, Y, Z)$ or by one of the couples $(F_1, F_2), (F_1^C, F_2^C)$.

When considering the fault dynamics, it is possible to identify four dynamic modes, each one described by a different system of autonomous ODEs: a sticking mode (00), corresponding to stationary asperities, and three slipping modes, associated with

the slip of asperity 1 alone (mode 10), the slip of asperity 2 alone (mode 01) and the simultaneous slip of the asperities (mode 11). A seismic event generally consists in $n$ slipping modes and involves one or both the asperities.

## 2.1 The sticking region

The sticking region of the system is defined as the set of states in which both asperities are stationary. During a global stick phase (mode 00), the rates $\dot{X}, \dot{Y}$ and $\dot{Z}$ are negligible with respect to their values when the asperities are slipping; thus, the



sticking region is a subset of the space $XYZ$.

The slip of asperity 1 occurs when

$$F_1 = -1, \tag{6}$$

5  while the slip of asperity 2 takes place when

$$F_2 = -\beta. \tag{7}$$

Combining these conditions with the expressions (2) of the forces, we obtain two planes in the $XYZ$ space,

$$X - \alpha Z - 1 = 0 \tag{8}$$

10  $$Y + \alpha Z - \beta = 0, \tag{9}$$

which we name $\Pi_1$ and $\Pi_2$, respectively. Of course, the Coulomb forces $F_1^C$ and $F_2^C$ vanish on $\Pi_1$ and $\Pi_2$, respectively; furthermore, their gradients

$$\nabla F_1^C = (1, 0, -\alpha), \qquad \nabla F_2^C = (0, 1, \alpha) \tag{10}$$

are orthogonal to $\Pi_1$ and $\Pi_2$, respectively.

We exclude overshooting during the slipping modes: accordingly, we assume $X \geq 0$ and $Y \geq 0$. As a consequence, the tangential forces on the asperities must always be in the same direction as the velocity of tectonic plates, i.e. $F_1 \leq 0$ and $F_2 \leq 0$. From Eq. (2), the limit cases $F_1 = 0$ and $F_2 = 0$ correspond to two planes in the $XYZ$ space,

$$X - \alpha Z = 0 \tag{11}$$

$$Y + \alpha Z = 0, \tag{12}$$

which we name $\Gamma_1$ and $\Gamma_2$, respectively.

To sum up, the sticking region of the system is the subset of the $XYZ$ space enclosed by the planes $X = 0, Y = 0, \Gamma_1, \Gamma_2, \Pi_1$
25  and $\Pi_2$: a convex hexahedron $\mathbf{H}$. Its vertices are the origin $(0, 0, 0)$ and the points

$$A = \left(0, 1, -\frac{1}{\alpha}\right), \ B = \left(\beta, 0, \frac{\beta}{\alpha}\right), \ C = \left(\beta + 1, 0, \frac{\beta}{\alpha}\right) \tag{13}$$



$$D = \left(0, \beta+1, -\frac{1}{\alpha}\right), \ E = (1,0,0), \ F = (0,\beta,0). \tag{14}$$

The sticking region is shown in Fig. (2) for a particular choice of the parameters $\alpha$ and $\beta$. Its volume can be expressed as a function of the parameters of the system as $\beta(\beta+1)/2\alpha$. Accordingly, the subset of the state space corresponding to stationary asperities decreases with the degree of coupling between the asperities and with the asymmetry of the system ($\beta \to 0$). By definition, every orbit of mode 00 is enclosed within $\mathbf{H}$ and eventually reaches one of its faces $AECD$ or $BCDF$, belonging to the planes $\Pi_1$ and $\Pi_2$, respectively, where an earthquake starts. In these cases, the system switches from mode 00 to mode 10 or mode 01, respectively. In the particular case in which the orbit of mode 00 reaches the edge $CD$, the system passes from mode 00 to mode 11.

## 3 Dynamic modes and slip in a seismic event

Let $P_0 \in \mathbf{H}$ be the state of the system at the beginning of an interseismic interval. The specific location of $P_0$ inside the sticking region allows the prediction of the first slipping mode involved in the next seismic event on the fault. In fact, Dragoni and Lorenzano (2015) illustrated the existence of a transcendental surface $\mathbf{\Sigma}$ within $\mathbf{H}$, expressed by the equation

$$V\Theta\left[W(\gamma_1) - W(\gamma_2)\right] + Y - X + 1 - \beta = 0, \tag{15}$$

where $W$ is the Lambert function with arguments

$$\gamma_1 = \frac{\alpha Z}{V\Theta}e^{-\frac{1-X}{V\Theta}}, \qquad \gamma_2 = -\frac{\alpha Z}{V\Theta}e^{-\frac{\beta-Y}{V\Theta}}. \tag{16}$$

The surface $\mathbf{\Sigma}$ divides $\mathbf{H}$ in two subsets $\mathbf{H_1}$ and $\mathbf{H_2}$ (Fig. 3). The seismic event starts with mode 10 if $P_0 \in \mathbf{H_1}$ or with mode 01 if $P_0 \in \mathbf{H_2}$; in the particular case in which $P_0 \in \mathbf{\Sigma}$, the seismic event starts with mode 11.

Mode 00 terminates at a point $P_1$ on the face $AECD$ or $BCDF$ of $\mathbf{H}$. The number and sequence of slipping modes involved in the subsequent seismic event can be discriminated from the specific position of $P_1$. If we consider the face $AECD$ (Fig. 4), the earthquake will be a one-mode event 10 if $P_1$ belongs to the trapezoid $\mathbf{Q_1}$; it will be a two-mode event 10-01 if $P_1$ belongs to the segment $\mathbf{s_1}$; it will be a three-mode event 10-11-01 or 10-11-10 if $P_1$ belongs to the trapezoid $\mathbf{R_1}$, where the precise sequence must be evaluated numerically and depends on the particular combination of the parameters $\alpha, \beta, \gamma$ and $\epsilon$. The remaining portion of the face would lead to overshooting. Analogous considerations can be made for the subsets $\mathbf{Q_2}, \mathbf{s_2}$ and $\mathbf{R_2}$ on the face $BCDF$. In the particular case in which $P_1$ belongs to the edge $CD$, the earthquake will be a two-mode event 11-01.

In addition, the knowledge of the position of $P_1$ allows to establish the total amount of slip of the asperities and the seismic moment associated with the earthquake. Let us consider an event made up of $n$ dynamic modes and let $P_i = (X_i, Y_i, Z_i)$ be the state of the system at time $T = T_i$, when the $i-$th mode starts ($i = 1, 2, ... n$). The final slip amplitudes of asperity 1 and 2 are,





respectively,

$$U_1 = X_1 - X_{n+1}, \qquad U_2 = Y_1 - Y_{n+1}. \tag{17}$$

Accordingly, the final seismic moment can be calculated as

$$M_0 = M_1 \frac{U_1 + U_2}{U}, \tag{18}$$

where $M_1$ and $U$ are the seismic moment and slip amplitude associated with a one-mode event 10 in the limit case $\gamma = 0$, respectively, with

$$U = 2\frac{1-\epsilon}{1+\alpha}. \tag{19}$$

The possible values of $U_1, U_2$ and $M_0$ are summarized in Table 1: the effect of wave radiation is characterized by means of the quantity

$$\kappa = \frac{1}{2}\left(1 + e^{-\frac{\gamma T_s}{2}}\right) \tag{20}$$

where $T_s$ is the duration of slip in a one-mode event (Dragoni and Santini, 2015).

As for the evolution of the variable $Z(T)$ during the earthquake, it changes according to the equation $\ddot{Z} = \ddot{Y} - \ddot{X}$, since the relaxation process is negligible during the slip of the asperities.

## 4 Stress perturbations from neighbouring faults

We now consider the perturbations of the state of the fault caused by the coseismic slip on surrounding faults. Following Dragoni and Piombo (2015), we assume that: (1) the perturbations occur during an interseismic interval; (2) the stress transfer takes place over a time interval negligible with respect to the duration of the interseismic interval; (3) at the time of the perturbation, the state of the fault is sufficiently far from the failure conditions and the stress transfer is small enough that the onset of motion of either asperity is not achieved immediately.

Let $(X, Y, Z) \in \mathbf{H}$ be the state of the fault at the time of the perturbation. Generally speaking, the system undergoes a transition to a new state

$$(X', Y', Z') = (X, Y, Z) + (\Delta X, \Delta Y, \Delta Z). \tag{21}$$

Since the stress transfer takes place over a time interval short with respect to the interseismic interval (assumption 2), viscoelastic relaxation is negligible during the perturbation and the rheology can be reasonably considered as purely elastic as the perturbation takes place. Accordingly, we set

$$\Delta Z = \Delta Y - \Delta X. \tag{22}$$



The change of state is then associated with a vector in the $XYZ$ space,

$$\Delta \mathbf{R} = (\Delta X, \Delta Y, \Delta Z).$$ (23)

The components of $\Delta \mathbf{R}$ generally have different magnitudes and may have different signs, as a consequence of the inhomogeneity of the stress field produced by an earthquake. They can be written in terms of the tangential forces $\Delta F_1$ and $\Delta F_2$
exerted by the perturbing source on asperity 1 and 2, respectively: from Eq. (2), we have

$$\Delta F_1 = -\Delta X + \alpha \Delta Z = \alpha \Delta Y - (1 + \alpha)\Delta X$$ (24)

$$\Delta F_2 = -\Delta Y - \alpha \Delta Z = \alpha \Delta X - (1 + \alpha)\Delta Y.$$ (25)

Combining these expressions together, we get

$$\Delta X = -\frac{1 + \alpha}{1 + 2\alpha}\Delta F_1 - \frac{\alpha}{1 + 2\alpha}\Delta F_2$$ (26)

$$\Delta Y = -\frac{\alpha}{1 + 2\alpha}\Delta F_1 - \frac{1 + \alpha}{1 + 2\alpha}\Delta F_2$$ (27)

$$\Delta Z = \frac{1}{1 + 2\alpha}\left(\Delta F_1 - \Delta F_2\right).$$ (28)

We conclude that the variations in tangential stress alter the orbit of the system.

The components of $\Delta \mathbf{R}$ can also be related to the orientation of the vector in the state space. With reference to Fig. (5), we have

$$\Delta X = \Delta R \cos \delta \cos \theta, \qquad \Delta Y = \Delta R \cos \delta \sin \theta, \qquad \Delta Z = \Delta R \sin \delta.$$ (29)

Introducing the assumption (22), the angle $\delta$ may be expressed in terms of the angle $\theta$ as

$$\delta = \arctan\left(\sin \theta - \cos \theta\right).$$ (30)

In writing Eq. (30), we took into account that

$$\delta \neq \frac{\pi}{2}, \frac{3\pi}{2}$$ (31)

or it would result

$$\Delta Z = \pm \Delta R, \quad \Delta X = \Delta Y = 0$$ (32)





which is a meaningless circumstance. From Eq. (29), the tangential forces (24)-(25) can be rewritten as

$$\Delta F_1 = \frac{\alpha \sin\theta - (1+\alpha)\cos\theta}{\sqrt{2 - \sin 2\theta}} \Delta R \tag{33}$$

$$\Delta F_2 = \frac{\alpha \cos\theta - (1+\alpha)\sin\theta}{\sqrt{2 - \sin 2\theta}} \Delta R. \tag{34}$$

Following the variations in normal stress, the static and dynamic frictions on each asperity are altered. Letting $f'_{s1}$ and $f'_{s2}$ be the new static frictions on asperity 1 and 2, respectively, we define

$$\beta_1 = \frac{f'_{s1}}{f_{s1}}, \qquad \beta_2 = \frac{f'_{s2}}{f_{s1}}. \tag{35}$$

The changes in static frictions are then

$$\Delta\beta_1 = \beta_1 - 1, \qquad \Delta\beta_2 = \beta_2 - \beta \tag{36}$$

on asperity 1 and 2, respectively.

Since the stress perturbation does not alter the friction coefficients of rocks, it is reasonable to assume that the ratio $\epsilon$ between dynamic and static friction is unchanged on both asperities. Therefore, letting $f'_{d1}$ and $f'_{d2}$ be the new dynamic frictions on asperity 1 and 2, respectively, we have

$$\frac{f'_{d1}}{f_{s1}} = \epsilon\frac{f'_{s1}}{f_{s1}} = \epsilon\beta_1, \qquad \frac{f'_{d2}}{f_{s1}} = \epsilon\frac{f'_{s2}}{f_{s1}} = \epsilon\beta_2. \tag{37}$$

The consequent changes in dynamic frictions are $\epsilon\Delta\beta_1$ and $\epsilon\Delta\beta_2$ on asperity 1 and 2, respectively.

### 4.1  Effects of the perturbation

The stress transfer resulting from earthquakes on neighbouring faults alters several parameters of the model. A first remarkable change concerns the strength of the asperities. After the perturbation, we can define a new ratio

$$\beta' = \frac{f'_{s2}}{f'_{s1}} = \frac{f'_{d2}}{f'_{d1}} = \frac{\beta_2}{\beta_1} \tag{38}$$

which differs from the original value $\beta$ given in Eq. (1). Moreover, the stress transfer may be so intense that the weaker asperity may become the stronger one: that is, it may result $\beta' > 1$.

The variations in static frictions entail different conditions for the onset of motion of the asperities. Taking Eq. (35) into

account, Eq. (6) and Eq. (7) become, respectively,

$$F_1 = -\beta_1, \qquad F_2 = -\beta_2. \tag{39}$$





By combination with Eq. (2), these conditions define the planes

$$X - \alpha Z - \beta_1 = 0 \tag{40}$$

$$Y + \alpha Z - \beta_2 = 0 \tag{41}$$

that we call $\Pi'_1$ and $\Pi'_2$, respectively. Conversely, the planes $\Gamma_1$ and $\Gamma_2$ are not affected by the stress perturbation, since they do not depend on frictions. We conclude that changes in normal stress modify the sticking region of the system, describing a new hexahedron $\mathbf{H}'$ in the state space. The coordinates of its vertices are

$$A' = \left(0, \beta_1, -\frac{\beta_1}{\alpha}\right), \; B' = \left(\beta_2, 0, \frac{\beta_2}{\alpha}\right), \; C' = \left(\beta_1 + \beta_2, 0, \frac{\beta_2}{\alpha}\right) \tag{42}$$

$$D' = \left(0, \beta_1 + \beta_2, -\frac{\beta_1}{\alpha}\right), \; E' = (\beta_1, 0, 0), \; F' = (0, \beta_2, 0). \tag{43}$$

The volume of $\mathbf{H}'$ is $\beta_1\beta_2(\beta_1 + \beta_2)/2\alpha$: thus, the set of states corresponding to stationary asperities is enlarged or reduced, depending on how normal stresses on the asperities are modified.

Following the changes in static frictions, the surface $\mathbf{\Sigma}$ in Eq. (15) is replaced by a new surface $\mathbf{\Sigma}'$ expressed by

$$V\Theta\left[W\left(\gamma'_1\right) - W\left(\gamma'_2\right)\right] + Y - X + \beta_1 - \beta_2 = 0, \tag{44}$$

where

$$\gamma'_1 = \frac{\alpha Z}{V\Theta}e^{-\frac{\beta_1 - X}{V\Theta}}, \qquad \gamma'_2 = -\frac{\alpha Z}{V\Theta}e^{-\frac{\beta_2 - Y}{V\Theta}}. \tag{45}$$

As a result, the sticking region $\mathbf{H}'$ is split in two subsets $\mathbf{H}'_1$ and $\mathbf{H}'_2$; furthermore, its faces $A'E'C'D'$ and $B'C'D'F'$ are divided into subsets $\mathbf{Q}'_1, \mathbf{s}'_1, \mathbf{R}'_1$ and $\mathbf{Q}'_2, \mathbf{s}'_2, \mathbf{R}'_2$, respectively.

As a consequence of the changes in dynamic frictions, the amount of slip that asperities undergo during a seismic event is modified. In turn, the perturbation alters the seismic moment associated with an earthquake. The variations in the final slip amplitudes $U_1$ and $U_2$ of asperity 1 and asperity 2, respectively, and in the final seismic moment $M_0$ associated with the different seismic events predicted by the model are listed in Table 2.

### 4.1.1   Changes in Coulomb forces

The variations in tangential stresses and static frictions discussed so far entail a change in the Coulomb forces assigned to the asperities. Combining Eq. (4) with Eq. (24) and Eq. (25), these changes are given by

$$\Delta F_1^C = -\Delta F_1 - \Delta\beta_1 = (1 + \alpha)\Delta X - \alpha\Delta Y - \Delta\beta_1 \tag{46}$$





$$\Delta F_2^C = -\Delta F_2 - \Delta\beta_2 = (1+\alpha)\Delta Y - \alpha\Delta X - \Delta\beta_2 \tag{47}$$

or, exploiting Eq. (33) and Eq. (34),

$$\Delta F_1^C = \frac{(1+\alpha)\cos\theta - \alpha\sin\theta}{\sqrt{2-\sin 2\theta}}\Delta R - \Delta\beta_1 \tag{48}$$

$$\Delta F_2^C = \frac{(1+\alpha)\sin\theta - \alpha\cos\theta}{\sqrt{2-\sin 2\theta}}\Delta R - \Delta\beta_2. \tag{49}$$

The sign of $\Delta F_i^C$ ($i=1,2$) determines whether the perturbation brings an asperity closer to or farther from the failure; specifically, positive variations entail that slip if favoured, and vice-versa. Equations (48) and (49) clearly point out that this effect is regulated by the orientation of the vector $\Delta\mathbf{R}$ in the state space. Bearing in mind the observations made in section 2.1, we find

that: $\Delta F_1^C$ is maximum when $\Delta\mathbf{R}$ is perpendicular to plane $\Pi_1$ and points toward it; it vanishes when $\Delta\mathbf{R}$ is parallel to plane $\Pi_1$; it is minimum when $\Delta\mathbf{R}$ is perpendicular to plane $\Pi_1$ and points away from it. Analogous considerations can be made for $\Delta F_2^C$.

On the whole, the effect of the stress perturbation can be discussed in terms of the quantity

$$\Delta F^C = \Delta F_2^C - \Delta F_1^C = (1+2\alpha)(\Delta Y - \Delta X) + \Delta\beta_1 - \Delta\beta_2. \tag{50}$$

Let us assume that the system is at a certain state $(X,Y,Z) \in \mathbf{H}_1$ before the perturbation; accordingly, the next seismic event on the fault will start with the failure of asperity 1. If $\Delta F^C > 0$, the perturbation favours the slip of asperity 2 more than the slip of asperity 1: therefore, the system is brought to a state closer to the condition for the simultaneous failure of the asperities and thus to the $\mathbf{\Sigma}$ surface. On the contrary, perturbations for which $\Delta F^C < 0$ take the system farther from the $\mathbf{\Sigma}$ surface. The opposite holds for an unperturbed state $(X,Y,Z) \in \mathbf{H}_2$.

### 4.1.2 Changes in the duration of the interseismic interval

As already stated, stress perturbations can anticipate or delay the occurrence of an earthquake produced by a certain asperity. We now quantify this effect in terms of the variation in the duration of the interseismic interval. Generally speaking, the perturbation vector $\Delta\mathbf{R}$ may cross the $\mathbf{\Sigma}$ surface and thus bring the system from an unperturbed state within $\mathbf{H}_1$ ($\mathbf{H}_2$) to a perturbed state within $\mathbf{H}_2'$ ($\mathbf{H}_1'$). For the sake of simplicity, we consider here only the particular case in which the perturbation vector

$\Delta\mathbf{R}$ does not cross the $\mathbf{\Sigma}$ surface. An example of a more general case will be shown in section 5 for a real fault.

Let us first focus on the case in which the unperturbed state $(X,Y,Z) \in \mathbf{H}_1$. The time required by the orbit of the system to reach plane $\Pi_1$, triggering the failure of asperity 1, was calculated by Amendola and Dragoni (2013) as

$$T_1 = \Theta W(\gamma_1) + \frac{1-X}{V} \tag{51}$$





with $\gamma_1$ given in Eq. (16). If the stress perturbation brings the system to a state $(X', Y', Z') \in \mathbf{H}'_1$ and the static friction on asperity 1 to $\beta_1$, the time required by the orbit to reach plane $\Pi'_1$ is

$$T'_1 = \Theta W(\gamma'_1) + \frac{\beta_1 - X'}{V} \qquad (52)$$

with $\gamma'_1$ given in Eq. (45). The difference between the two times is

$$\Delta T_1 = T'_1 - T_1 = \Theta[W(\gamma'_1) - W(\gamma_1)] - \frac{\Delta F_1^C + \alpha \Delta Z}{V} \qquad (53)$$

where Eq. (46) has been employed.

If instead $(X, Y, Z) \in \mathbf{H}_2$, the time required by the orbit of the system to reach plane $\Pi_2$, triggering the failure of asperity 2, is given by (Amendola and Dragoni, 2013)

$$T_2 = \Theta W(\gamma_2) + \frac{\beta - Y}{V} \qquad (54)$$

with $\gamma_2$ given in Eq. (16). If the stress perturbation takes the system to a state $(X', Y', Z') \in \mathbf{H}'_2$ and the static friction on asperity 2 to $\beta_2$, the time required to reach plane $\Pi'_2$ is

$$T'_2 = \Theta W(\gamma'_2) + \frac{\beta_2 - Y'}{V} \qquad (55)$$

with $\gamma'_2$ given in Eq. (45). The difference between the two times is

$$\Delta T_2 = T'_2 - T_2 = \Theta[W(\gamma'_2) - W(\gamma_2)] - \frac{\Delta F_2^C - \alpha \Delta Z}{V} \qquad (56)$$

where Eq. (47) has been employed. Positive values of $\Delta T_1$ and $\Delta T_2$ correspond to a delay in the occurrence of an earthquake on asperity 1 and 2, respectively, and vice-versa.

### 4.1.3 Discussion

According to the model, rock rheology plays a critical role in the response to stress perturbations. In the case of purely elastic coupling between the asperities, Dragoni and Piombo (2015) showed that the changes in the duration of the interseismic interval prior to the failure of asperity 1 and 2 are, respectively,

$$\Delta T_1 = -\frac{\Delta F_1^C}{V}, \qquad \Delta T_2 = -\frac{\Delta F_2^C}{V} \qquad (57)$$

Accordingly, an increase in the Coulomb force associated with a given asperity ($\Delta F_i^C > 0$) directly yields the anticipation of the slip of that asperity, and vice-versa. What is more, the variation in the duration of the interseismic interval is proportional to the change in the Coulomb force associated with the asperity.

Conversely, in the viscoelastic case there is no straightforward connection between the sign of $\Delta F_i^C$ and the anticipation or delay of an earthquake on the associated asperity. In fact, the expressions (53) and (56) obtained for $\Delta T_1$ and $\Delta T_2$ in the previous section indicate that the net effect depends in a non trivial way on the particular state of the fault at the time of the stress perturbation and right after it. This result points out the complex interplay between the post-seismic evolution of a fault in the presence of viscoelastic relaxation and the stress transfer from neighbouring faults.





## 5 An application: perturbation of the 1992 Landers fault by the 1999 Hector Mine earthquake

We study the effects of the 16 October 1999 $M_w\,7.1$ Hector Mine, California, earthquake on the post-seismic evolution of the fault that generated the 28 June 1992 $M_w\,7.3$ Landers, California, earthquake. The geometry of the two faults is shown in Fig. (6).

The 1992 Landers earthquake was a right-lateral strike-slip event that can be approximated as the result of the slip of two coplanar asperities (Kanamori et al., 1992): a northern one (asperity 1) and a southern one (asperity 2), with average slips $u_1 = 6\,\text{m}$ and $u_2 = 3\,\text{m}$, respectively. Following Dragoni and Tallarico (2016), we assume a common area $A = 300\,\text{km}^2$ for both the asperities. We place the centres of asperity 1 and asperity 2 at (34.46° N, 116.52° W) and (34.20° N, 116.44° W), respectively, with a common depth of $8$ km. The earthquake initiated with the failure of asperity 2, followed by the failure of asperity 1. We characterize the event by strike, dip and rake angles of $345°, 85°$ and $180°$, respectively, an average of the values provided by Kanamori et al. (1992) for the two phases of the earthquake.

The 1992 event was followed by remarkable post-seismic deformation, which can be interpreted as the result of several processes. For the sake of the present application, we assume viscoelastic relaxation as the most significant mechanism. We assign a viscosity $\eta = 5 \cdot 10^{18}\,\text{Pa\,s}$ to the lower crust at Landers, averaging the estimates provided by Deng et al. (1998), Pollitz et al. (2000), Freed and Lin (2001) and Masterlark and Wang (2002). With a rigidity $\mu = 30\,\text{GPa}$, the corresponding Maxwell relaxation time is $\tau = \eta/\mu \simeq 5\,\text{a}$.

We model the 1992 earthquake as a two-mode event 01-10 starting from mode 00. Accordingly, the orbit of the system during mode 00 lies inside the subset $\mathbf{H_2}$ of the sticking region and the state $P_1$ at the beginning of the earthquake belongs to segment $\mathbf{s_2}$ (Fig. 4). The coordinates of $P_1$ are

$$X_1 = \alpha Z_1 + 1 - \alpha\beta\kappa U, \quad Y_1 = \beta - \alpha Z_1, \quad Z_1 \tag{58}$$

with

$$Z_a \le Z_1 \le Z_b, \tag{59}$$

where the extreme values $Z_a$ and $Z_b$ correspond to the end points of $\mathbf{s_2}$:

$$Z_a = \frac{\kappa U(\alpha\beta + 1) - 1}{\alpha}, \qquad Z_b = \frac{\beta(1 - \kappa U)}{\alpha}. \tag{60}$$

At the end of mode 01, the system is at point $P_2$ with coordinates

$$X_2 = X_1, \quad Y_2 = Y_1 - \beta\kappa U, \quad Z_2 = Z_1 - \beta\kappa U, \tag{61}$$

where mode 10 starts. As $Z_1$ varies in the interval given in Eq. (59), an infinite number of points $P_2$ describe a segment $\mathbf{r_2}$ on the subset $\mathbf{Q_1}$ of the face $AECD$ and parallel to the edge $CD$. Mode 10 terminates at point $P_3$ with coordinates

$$X_3 = X_2 - \kappa U, \quad Y_3 = Y_2, \quad Z_3 = Z_2 + \kappa U. \tag{62}$$





Again, as $Z_1$ varies in the interval given in Eq. (59), there is an infinite number of points $P_3$ defining another segment $\mathbf{q_2}$ parallel to the edge $CD$. This segment is situated within the sticking region and crosses the surface $\mathbf{\Sigma}$ for $Z_1 = Z_c$, with $Z_a < Z_c < Z_b$.

Dragoni and Tallarico (2016) studied the 1992 Landers earthquake under the hypothesis of purely elastic coupling between
the asperities. Following the authors, we take $\alpha = 0.1$, $\beta = 0.5$, $\gamma = 1.5$ and $\epsilon = 0.7$, a set of values yielding modelled moment rate and seismic spectrum comparable with the observations. Thus, we have $U \simeq 0.546$ and $\kappa \simeq 0.52$. As for viscoelastic relaxation, it can be characterized by the product $V\Theta$ (Amendola and Dragoni, 2013), which can be estimated as

$$V\Theta = \frac{\kappa U v \tau}{u_1}, \tag{63}$$

where $v = 3\,\mathrm{cm\,a^{-1}}$ is the relative plate velocity at Landers (Wallace, 1990). Accordingly, we have $V\Theta \simeq 0.007$.

Every state $P_1$ on segment $\mathbf{s_2}$, where the 1992 earthquake begun, corresponds to a specific state $P_3$ on segment $\mathbf{q_2}$, where the 1992 earthquake ended. Exploiting Eq. (61), we can express the coordinates (62) of $P_3$ as a function of $Z_1$. Since $\mathbf{q_2}$ crosses the surface $\mathbf{\Sigma}$, the state $P_3$ can belong to $\mathbf{H_1}, \mathbf{H_2}$ or $\mathbf{\Sigma}$, in correspondence to $Z_c < Z_1 \leq Z_b$, $Z_a \leq Z_1 < Z_c$ and $Z_1 = Z_c$, respectively. In the first case, the next event will start with the failure of asperity 1; in the second case, with the failure of
asperity 2; in the third case, with the simultaneous failure of the asperities. With the values of $\alpha, \beta, \kappa$ and $U$ listed above, we find $Z_a \simeq -7.02$, $Z_b \simeq 3.58$ and $Z_c \simeq 0.78$. Accordingly, only about one fourth of segment $\mathbf{q_2}$ lies inside the subset $\mathbf{H_1}$ of the sticking region. Without any further discussion and neglecting the stress perturbation caused by the Hector Mine earthquake, we would infer that future events on the 1992 fault are more likely to start with the failure of asperity 2.

## 5.1 Stress perturbation by the 1999 Hector Mine earthquake

The 1999 Hector Mine earthquake was generated by right-lateral strike-slip faulting located at $(34.59°\,\mathrm{N}, 116.27°\,\mathrm{W})$, about 20 km northeast from the Landers fault (Jónsson et al., 2002; Salichon et al., 2004). We characterize the event averaging the data available in the SRCMOD database and assume: strike, dip and rake angles of $330°$, $80°$ and $180°$, respectively; a depth of 10 km; a seismic moment of $6.62 \cdot 10^{19}\,\mathrm{Nm}$.

The stress transferred to the asperities at Landers can be evaluated employing the model of Appendix A, taking

$$\phi_1 = 345°, \quad \phi_2 = 330°, \qquad \psi_1 = 85°, \quad \psi_2 = 80°, \qquad \lambda_1 = \lambda_2 = 180°. \tag{64}$$

As a result, the normal and tangential components of the perturbing stress on asperity 1 are

$$\sigma_{1n} \simeq 0.14\,\mathrm{MPa}, \qquad \sigma_{1t} \simeq 0.39\,\mathrm{MPa}. \tag{65}$$

Accordingly, the static friction on asperity 1 is reduced and right-lateral slip is favoured. As for asperity 2, the components of
the perturbing stress are

$$\sigma_{2n} \simeq 0.18\,\mathrm{MPa}, \qquad \sigma_{2t} \simeq -0.17\,\mathrm{MPa}, \tag{66}$$



suggesting that static friction on asperity 2 is reduced and right-lateral slip is inhibited.

We now introduce the effect of the perturbation in the framework of the discrete model. The changes in the tangential forces (2) on the asperities are

$$\Delta F_1 = -\frac{\sigma_{1t}}{f_{s1}}A, \qquad \Delta F_2 = -\frac{\sigma_{2t}}{f_{s1}}A. \tag{67}$$

The static friction $f_{s1}$ on asperity 1 can be evaluated as (Dragoni and Santini, 2012)

$$f_{s1} = \frac{Ku_1}{\kappa U} \tag{68}$$

where the constant

$$K = \frac{\mu A}{d} \tag{69}$$

is an expression of the coupling between the asperities and the tectonic plates. With $d = 80$ km (Masterlark and Wang, 2002), it results $f_{s1}/A \simeq 7.9$ MPa. Hence, we have

$$\Delta F_1 \simeq -0.05, \qquad \Delta F_2 \simeq 0.02. \tag{70}$$

From Eq. (26) – (28), the components of the perturbation vector $\Delta\mathbf{R}$ are

$$\Delta X \simeq 0.043, \qquad \Delta Y \simeq -0.016, \qquad \Delta Z \simeq -0.059. \tag{71}$$

As a result, the orientation of $\Delta\mathbf{R}$ in the state space is characterized by angles $\theta \simeq -0.35$ rad and $\delta \simeq -0.91$ rad. The changes in static frictions (36) can be calculated as

$$\Delta\beta_1 = -\frac{k_s\sigma_{1n}}{f_{s1}}A, \qquad \Delta\beta_2 = -\frac{k_s\sigma_{2n}}{f_{s1}}A, \tag{72}$$

where $k_s$ is the effective static friction coefficient on asperity 1. Assuming $k_s = 0.4$, we get

$$\Delta\beta_1 \simeq -0.0073, \qquad \Delta\beta_2 \simeq -0.0092. \tag{73}$$

Finally, from Eq. (46) and Eq. (47), the changes in Coulomb forces on the asperities are

$$\Delta F_1^C \simeq 0.057, \qquad \Delta F_2^C \simeq -0.012. \tag{74}$$

At the time of the Hector Mine earthquake, the Landers fault was at a state $P_4$ resulting from the post-seismic evolution of any of the possible states $P_3 \in \mathbf{q_2}$ where the 1992 event ended. The coordinates of $P_4$ can be calculated from the solution to the equations of mode 00 given by Dragoni and Lorenzano (2015) and taking into account that the time interval $\tilde{t}$ elapsed between the 1992 Landers and 1999 Hector Mine earthquakes amounts to about 7.3 years:

$$X_4 = X_3 + V\Theta\tilde{T}, \qquad Y_4 = Y_3 + V\Theta\tilde{T}, \qquad Z_4 = Z_3 e^{-\tilde{T}}, \tag{75}$$





where

$$\tilde{T} = \frac{\tilde{t}}{\tau} \approx 1.5. \tag{76}$$

Making use of Eq. (61) and Eq. (62), we can express the coordinates of $P_4$ as a function of $Z_1 \in [Z_a, Z_b]$. Accordingly, there is an infinite number of points $P_4$ defining a vector $\mathbf{t_2}$ inside the sticking region. At $T = \tilde{T}$, the perturbation vector $\Delta \mathbf{R}$ moves

every state $P_4$ to a new state $P_4'$ with coordinates

$$X_4' = X_4 + \Delta X, \qquad Y_4' = Y_4 + \Delta Y, \qquad Z_4' = Z_4 + \Delta Z \tag{77}$$

which can be expressed as a function of $Z_1 \in [Z_a, Z_b]$. As a result, a new vector $\mathbf{t_2'}$ identifies the state of the Landers fault after the Hector Mine earthquake.

In order to characterize the effect of the perturbation, let us consider the difference $\Delta F^C$ defined in Eq. (50): from Eq. (74), we get $\Delta F^C \simeq -0.069$. Since $\Delta F^C < 0$, we conclude that the stress perturbation is such that: states $P_4 \in \mathbf{H_1}$ are moved to $\mathbf{H_1'}$; the state $P_4 \in \mathbf{\Sigma}$ enters $\mathbf{H_1'}$; states $P_4 \in \mathbf{H_2}$ are shifted towards the $\mathbf{\Sigma}$ surface and some of them enter $\mathbf{H_1'}$. Specifically, we find that $P_4'$ belongs to $\mathbf{H_1'}$, $\mathbf{H_2'}$ and $\mathbf{\Sigma'}$ in correspondence to $Z_c' < Z_1 \leq Z_b$, $Z_a \leq Z_1 < Z_c'$ and $Z_1 = Z_c'$, with $Z_c' \simeq 0.50$. On the whole, we can draw the preliminary conclusion that the stress perturbation is such that future events on the Landers fault

starting with the slip of asperity 1 are favoured over events starting with the slip of asperity 2. A deeper discussion is provided in the following.

## 5.2    Constraints due to the seismic history to date

In order to improve our knowledge on the state that gave rise to the 1992 Landers earthquake and on the possible future events generated by that fault, we exploit the seismic history between 1999 and the present date. After the perturbation caused by the

Hector Mine earthquake, the interseismic time $T_{is}'$ of the Landers fault can be calculated from Eq. (52) and Eq. (55) for states $P_4'$ belonging to $\mathbf{H_1'}$ and $\mathbf{H_2'}$, respectively:

$$T_{is}' = \begin{cases} \Theta W(\gamma_1') + \frac{\beta_1 - X_4'}{V}, & Z_c' < Z_1 \leq Z_b \\ \Theta W(\gamma_2') + \frac{\beta_2 - Y_4'}{V}, & Z_a \leq Z_1 < Z_c' \end{cases}, \tag{78}$$

where

$$\gamma_1' = \frac{\alpha Z_4'}{V\Theta} e^{-\frac{\beta_1 - X_4'}{V\Theta}}, \qquad \gamma_2' = -\frac{\alpha Z_4'}{V\Theta} e^{-\frac{\beta_2 - Y_4'}{V\Theta}}. \tag{79}$$

Since no earthquakes have been produced by the Landers fault after the occurrence of the Hector Mine event, up to year 2016, we can exclude the states on the segment $\mathbf{s_2}$ yielding an expected interseismic time (78) shorter than or equal to $t_{is}' = 17$ years. The requirement

$$T_{is}' > \frac{t_{is}'}{\tau} \Theta \approx 3.5 \Theta \tag{80}$$





is satisfied by states on segment $\mathbf{s_2}$ in the subset $\tilde{Z}_a \le Z_1 \le \tilde{Z}_b$, with $\tilde{Z}_a \simeq -1.17$ and $\tilde{Z}_b \simeq 2.19$.

As a consequence, we can constrain the admissible states on the segment $\mathbf{t_2}$. A comparison between the intervals $[\tilde{Z}_a, Z_c]$ and $[Z_c, \tilde{Z}_b]$ points out that more than one half of the acceptable subset of $\mathbf{t_2}$ belongs to $\mathbf{H_2}$. Hence, before the stress perturba-
tion caused by the Hector Mine earthquake, future events on the 1992 Landers fault were more likely to start with the failure of asperity 2.

In turn, the refinement of $\mathbf{t_2}$ limits the acceptable states on the segment $\mathbf{t'_2}$. From the amplitude of the intervals $[\tilde{Z}_a, Z'_c]$ and $[Z'_c, \tilde{Z}_b]$, we deduce that the acceptable subset of $\mathbf{t'_2}$ is almost equally divided between $\mathbf{H'_1}$ and $\mathbf{H'_2}$. Therefore, if we consider the influence of the Hector Mine earthquake on future events generated by the 1992 Landers fault, we conclude that the stress
perturbation yielded homogenization in the probability of events starting with the failure of asperity 1 or asperity 2. This result is in agreement with the observation that the perturbation vector $\Delta \mathbf{R}$ shifted the whole segment $\mathbf{t_2}$ towards the subset $\mathbf{H'_1}$ of the sticking region.

These conclusions would have to be reconsidered if new stress perturbations from neighbouring faults were to affect the
post-seismic evolution of the Landers fault in the future. In addition, if no earthquakes were to be observed for some time on the Landers fault, the refining procedure discussed above could be repeated and the admissible subsets of segments $\mathbf{s_2}$, $\mathbf{t_2}$ and $\mathbf{t'_2}$ could be constrained with further precision.

### 5.3    Effects of the stress perturbation on future earthquakes

Finally, we discuss the features of the next seismic event generated by the 1992 Landers fault, highlighting the changes due to
the Hector Mine earthquake.

Every state $P_1 \in \mathbf{s_2}$ where the 1992 earthquake begun corresponds to a particular state $P_4 \in \mathbf{t_2}$ and $P'_4 \in \mathbf{t'_2}$ before and after the stress perturbation associated with the Hector Mine earthquake, respectively. Since the segment $\mathbf{t_2}$ intersects the surface $\Sigma$, the state $P_4$ can belong to $\mathbf{H_1}, \mathbf{H_2}$ or $\Sigma$ (Fig. 3), thus affecting the asperity that will fail the first at the beginning of the
next earthquake on the fault. In the first case, the next event will start with the failure of asperity 1, in the second case with the failure of asperity 2, in the third case with the simultaneous failure of the asperities. Analogous considerations hold for states $P'_4$ in $\mathbf{H'_1}, \mathbf{H'_2}$ and $\Sigma'$, respectively.

The number and the sequence of dynamic modes in the earthquake depend on the subinterval of $Z_1$ considered. The details
are summarized in Table 3 for both the unperturbed and perturbed cases. Taking these specifics into account and referring to Table 1 and Table 2, we evaluate the seismic moments $M_0$ and $M'_0$ associated with the expected future earthquake on the 1992 fault before and after the Hector Mine earthquake, respectively. In Fig. (7), we show the difference

$$\Delta M_0 = M'_0 - M_0 \tag{81}$$



as a function of $Z_1 \in [\tilde{Z}_a, \tilde{Z}_b]$. Owing to the translation imposed to the segment $\mathbf{t_2}$ by the perturbation vector $\Delta\mathbf{R}$, the sign of $\Delta M_0$ changes across the different subintervals of $Z_1$. The energy released by the earthquake is increased for $Z_1 \in [0.43, 0.71]$, while it is reduced elsewhere.

Another significant result of the stress perturbation concerns the variation in the interseismic time before the next seismic event. As in section 5.2, we consider the post-seismic evolution from 1999 onwards and set the origin of times at the occurence of the Hector Mine earthquake. The expected interseismic time $T_{is}$ prior to the stress perturbation can be calculated from Eq. (51) and Eq. (54) for states $P_4$ belonging to $\mathbf{H_1}$ and $\mathbf{H_2}$, respectively:

$$T_{is} = \begin{cases} \Theta W(\gamma_1) + \frac{1-X_4}{V}, & Z_c < Z_1 \leq \tilde{Z}_b \\ \Theta W(\gamma_2) + \frac{\beta-Y_4}{V}, & \tilde{Z}_a \leq Z_1 < Z_c \end{cases}, \tag{82}$$

where

$$\gamma_1 = \frac{\alpha Z_4}{V\Theta} e^{-\frac{1-X_4}{V\Theta}}, \qquad \gamma_2 = -\frac{\alpha Z_4}{V\Theta} e^{-\frac{\beta-Y_4}{V\Theta}}. \tag{83}$$

The interseismic time $T'_{is}$ after the stress perturbation has been given in Eq. (78). The difference

$$\Delta T = T'_{is} - T_{is} \tag{84}$$

is shown in Fig. (8) as a function of $Z_1 \in [\tilde{Z}_a, \tilde{Z}_b]$. For states $P_4 \in \mathbf{H_1}$ corresponding to $P'_4 \in \mathbf{H'_1}$ and states $P_4 \in \mathbf{H_2}$ corre-
sponding to $P'_4 \in \mathbf{H'_2}$, this difference coincides with (53) and (56), respectively.
Some peculiar features stand out. First, we notice that, for all states $P_4 \in \mathbf{H_2}$ corresponding to $P'_4 \in \mathbf{H'_2}$, that is, for $Z_1 \in [\tilde{Z}_a, Z'_c]$, the interseismic time is increased by the stress perturbation, in agreement with the inhibiting effect on asperity 2 suggested by Eq. (74). On the other hand, Eq. (74) suggests that the failure of asperity 1 is promoted, but this is not verified by all states $P'_4 \in \mathbf{H'_1}$, that is, for $Z_1 \in [Z'_c, \tilde{Z}_b]$. In fact, the interseismic time is reduced only for $Z_1 \in (0.53, \tilde{Z}_b]$, while it is
increased for $Z_1 \in [Z'_c, 0.53)$. In the particular case $Z_1 = 0.53$, there is no change in the interseismic time. This is a remarkable result, showing that the presence of viscoelastic relaxation at the time of the stress perturbation entails the unpredictability of the consequent influence in terms of anticipation/delay of future earthquakes, on the basis of the sole knowledge of the change in Coulomb stress.

At the occurrence of the next earthquake produced by the Landers fault, the number and sequence of dynamic modes involved and the energy released will reveal more about the state of the system, thus allowing a further refinement of the specific conditions that gave rise to the 1992 event.

## 6  Conclusions

We considered a plane fault embedded in a shear zone, subject to a uniform strain rate owing to tectonic loading. The fault is
characterized by the presence of two asperities with equal areas and different frictional resistance. The coseismic static stress



field due to earthquakes produced by the fault is relaxed by viscoelastic deformation.

The fault was treated as a discrete dynamical system with three degrees of freedom: the slip deficits of the asperities and the variation of their difference due to viscoelastic deformation. The dynamics of the system was described in terms of one sticking

mode and three slipping modes. In the sticking mode, the orbit of the system lies in a convex hexahedron in the space of the state variables, while the number and the sequence of slipping modes during a seismic event are determined by the particular state of the system at the beginning of the interseismic interval preceding the event. The amount of slip of the asperities and the energy released during an earthquake generated by the fault can be predicted accordingly.

The effect of stress transfer due to earthquakes on neighbouring faults was studied in terms of a perturbation vector yielding changes to the state of the system, its sticking region and the energy released during a subsequent seismic event. The specific effect on the evolution of the fault is related with the orientation of this vector in the state space.

We investigated the interplay between the ongoing viscoelastic relaxation on the fault and a stress perturbation imposed

during an interseismic interval. Following a stress perturbation, the anticipation/delay of the failure of one asperity is connected with the change in the associated Coulomb stress. In particular, the variation in the difference between the Coulomb stresses of the two asperities influences the possibility of their simultaneous slip during the next seismic event. However, the presence of viscoelastic relaxation prevents any prediction about the change in the interseismic time of the fault, which is conditioned by the particular state of the fault at the time of the stress perturbation and immediately after it. This result represents the most

significant difference with respect to the purely elastic case considered by Dragoni and Piombo (2015), where the advance or delay of an earthquake on the fault can be straightforwardly inferred from the sole knowledge of the change in Coulomb stress on the fault.

We applied the model to the stress perturbation imposed by the 1999 Hector Mine, California, earthquake to the fault that

originated the 1992 Landers, California, earthquake, which was due to the failure of two asperities and was followed by significant viscoelastic relaxation. We modelled the 1992 Landers earthquake as a two-mode event associated with the separate slip of the asperities and showed how the event is compatible with a number of possible initial states of the fault, which can be screened on the basis of the seismic history to date. The details of the stress transfer associated with the 1999 Hector Mine earthquake were calculated using the relative positions and faulting styles of the two faults as a starting point. We discussed the

effect of the stress perturbation, pointing out the complexity of its influence on the possible future events generated by the 1992 fault in terms of the associated energy release, the sequence of dynamic modes involved and the duration of the interseismic interval. On the whole, the application allowed to exemplify the critical unpredictability of the effect of a stress perturbation occurring while viscoelastic relaxation is taking place.





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

*Author contributions.*  E. L. developed the model, produced the figures and wrote a preliminary version of the paper; M. D. checked the
10 equations and revised the text. Both authors discussed extensively the results.

*Competing interests.*  No competing interests are present.





## List of Figure Captions

Fig. 1 - Sketch of the model of a plane fault with two asperities. The rectangular frame is the fault border. The state of the asperities is described by their slip deficits $X(T)$ and $Y(T)$, while the variable $Z(T)$ represents the variation of the difference between the slip deficits of the asperities due to viscoelastic deformation.

5 Fig. 2 - The sticking region of the system: a convex hexahedron $\mathbf{H}$ ($\alpha = 1, \beta = 1$).

Fig. 3 - The surface $\mathbf{\Sigma}$ that splits the sticking region $\mathbf{H}$ in two subsets $\mathbf{H_1}$ (below) and $\mathbf{H_2}$ (above) ($\alpha = 1, \beta = 1, V\Theta = 1$). It allows to discriminate the first slipping mode during an earthquake.

Fig. 4 - The subsets of the faces $AECD$ and $BCDF$ of the sticking region $\mathbf{H}$, regulating the number and sequence of slipping modes during a seismic event ($\alpha = 1, \beta = 1, \gamma = 1, \epsilon = 0.7$).

10 Fig. 5 - The vector $\mathbf{\Delta R}$ and its orientation in the $XYZ$ space, characterizing the stress perturbation imposed on the system by earthquakes produced by neighbouring faults.

Fig. 6 - Geometry of the Landers (LAN) and Hector Mine (HM), California, faults that generated the 1992 and 1999 earthquakes, respectively. The stars indicate the hypocentres of the seismic events. The labels 1 and 2 identify the asperities on the Landers fault.

15 Fig. 7 - Change in the seismic moment released during the next event on the 1992 Landers, California, fault, as a result of the stress perturbation due to the 1999 Hector Mine, California, earthquake. On the horizontal axis, the variable $Z_1$ describing the initial state of the 1992 event. The values $Z_1 = Z_c$ and $Z_1 = Z_c'$ correspond to the largest possible earthquakes predicted by the model before and after the stress perturbation, respectively, associated with a sequence of modes 11-01.

Fig. 8 - Change in the interseismic time before the next event on the 1992 Landers, California, fault, as a result of the stress 20 perturbation due to the 1999 Hector Mine, California, earthquake. On the horizontal axis, the variable $Z_1$ describing the initial state of the 1992 event. The values $Z_1 = Z_c$ and $Z_1 = Z_c'$ correspond to the largest possible earthquakes predicted by the model before and after the stress perturbation, respectively, associated with a sequence of modes 11-01.

Fig. A1 - Geometry of the model employed to study the stress transfer between neighbouring faults. Fault 1 is the perturbing fault, while fault 2 is the receiving fault. The coordinates $(E, N, D)$ are the UTM coordinates and depth of the centres of the 25 faults, respectively, whereas the axes $(x, y, z)$ correspond with the directions of dip, strike and normal on fault 1, respectively. The angles $\phi$ and $\psi$ are the strike and dip angles of the faults, respectively.





# Figures

**Figure 1.**

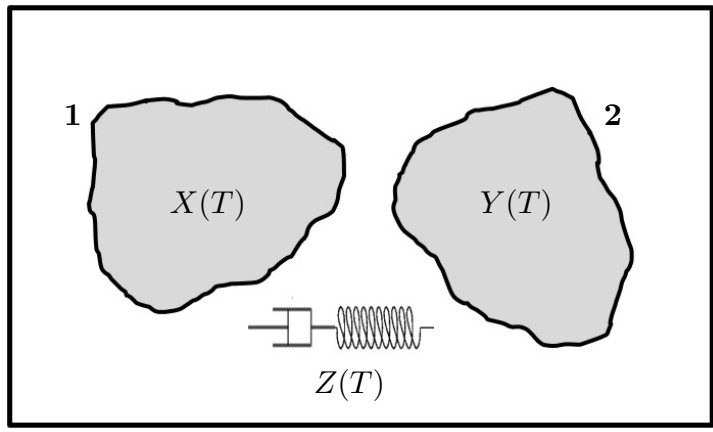

**Figure 2.**

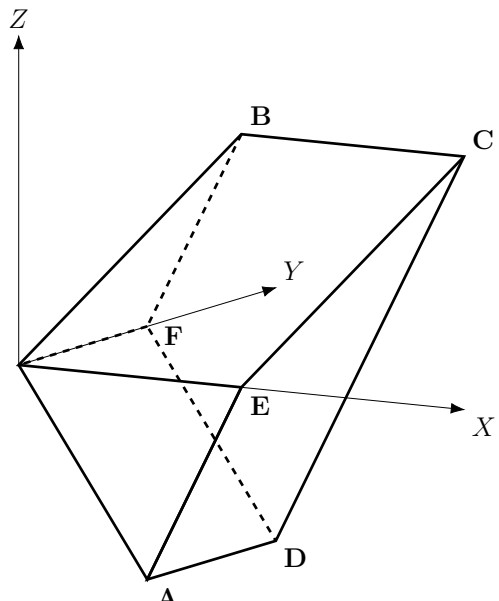

**Figure 3.**

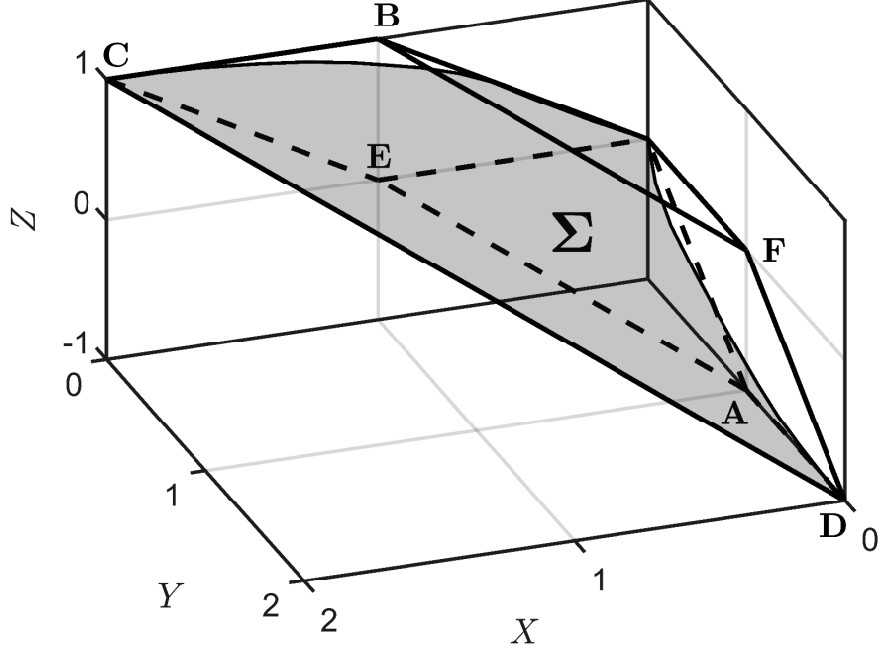

**Figure 4.**

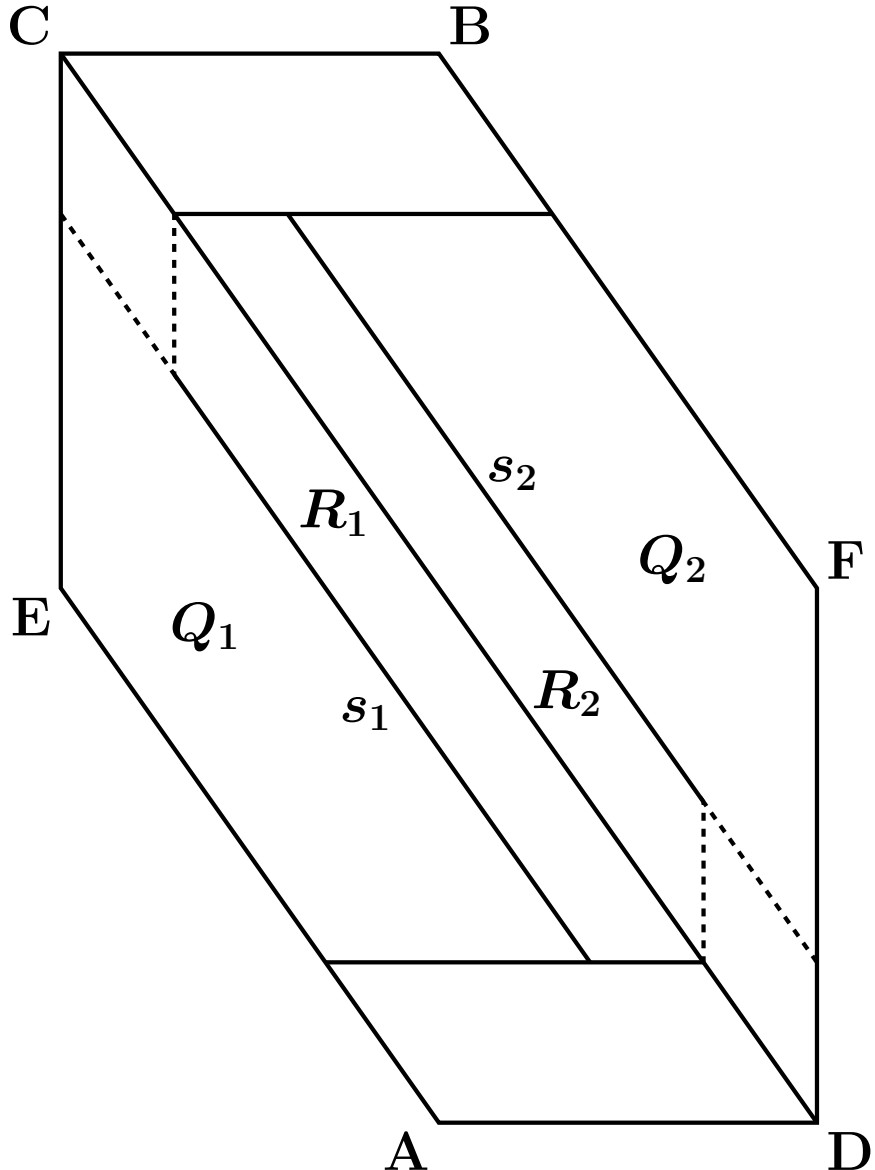



**Figure 5.**

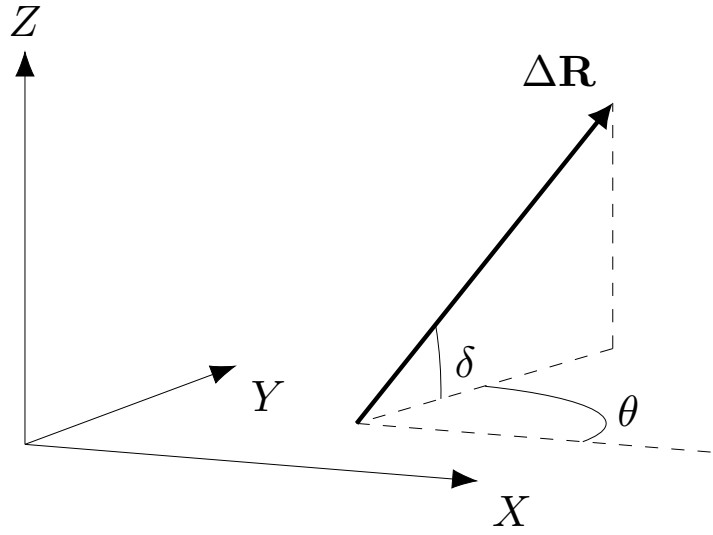

**Figure 6.**

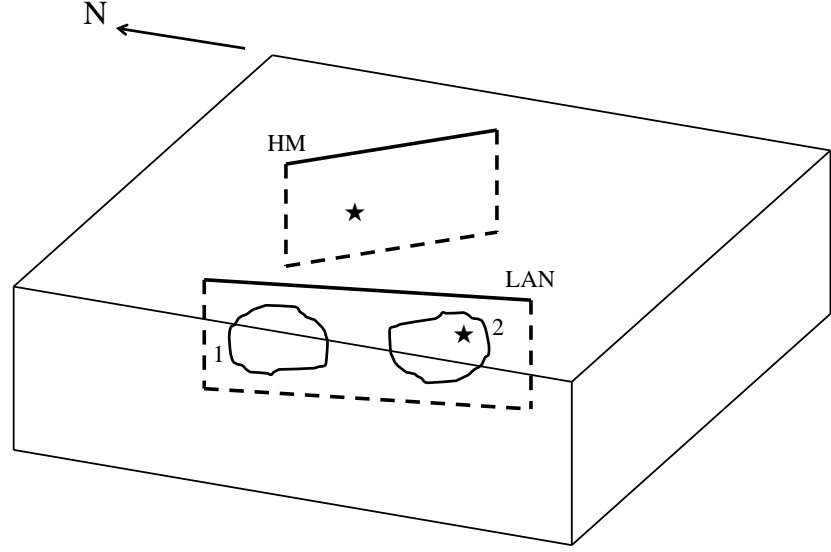




**Figure 7.**

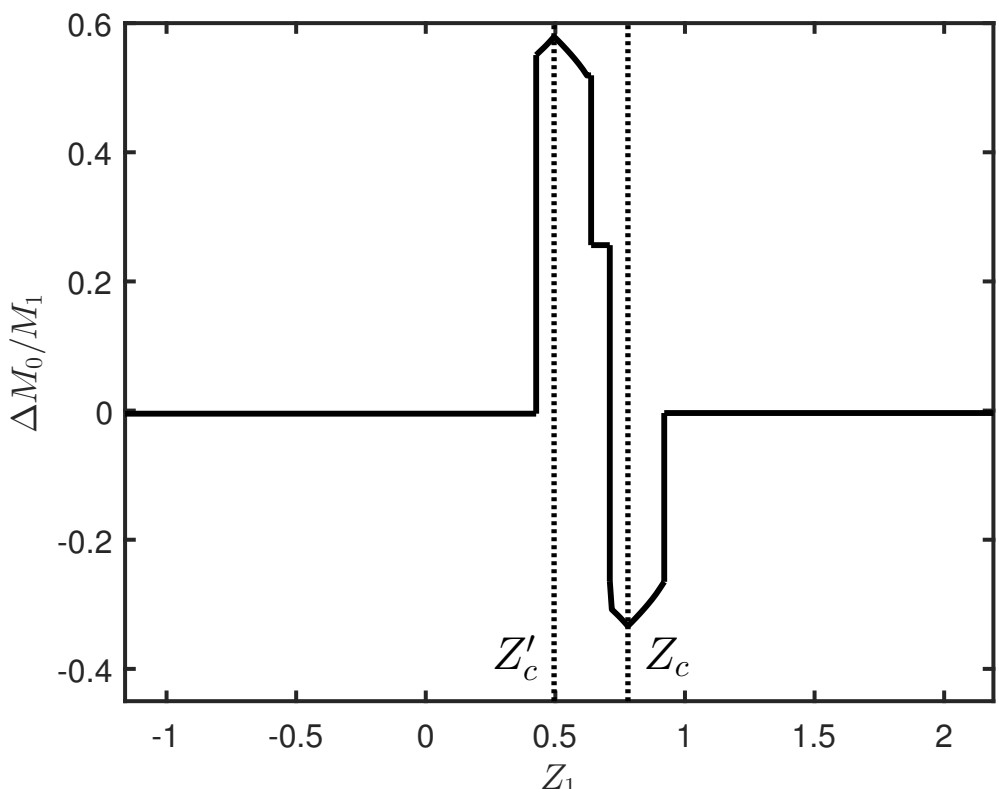



**Figure 8.**

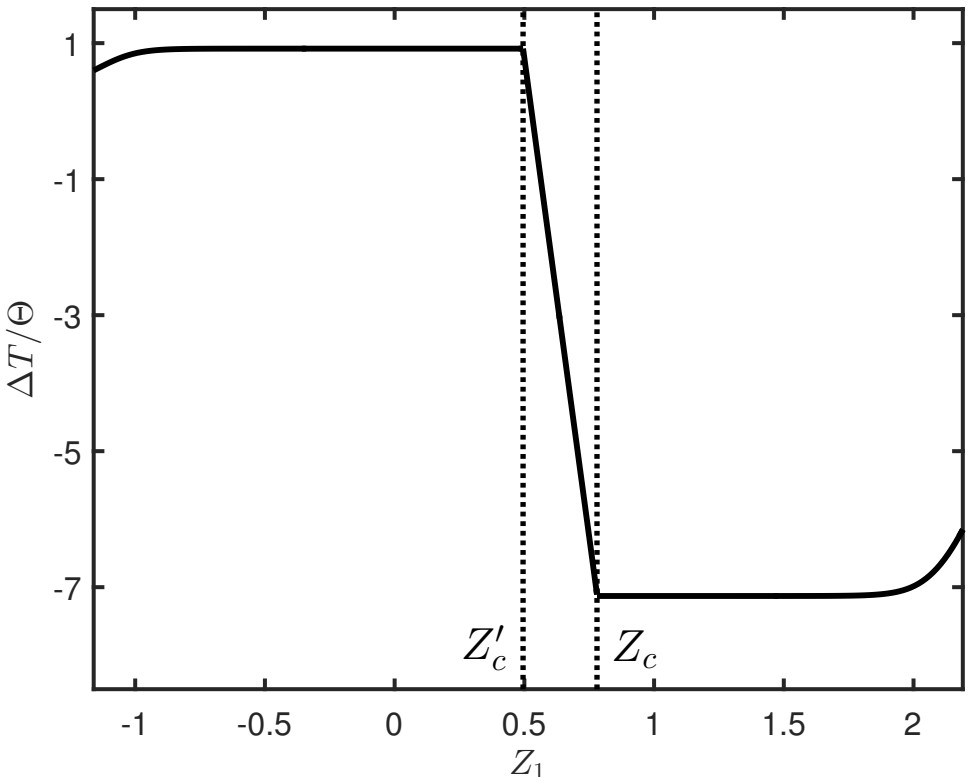

**Figure A1.**

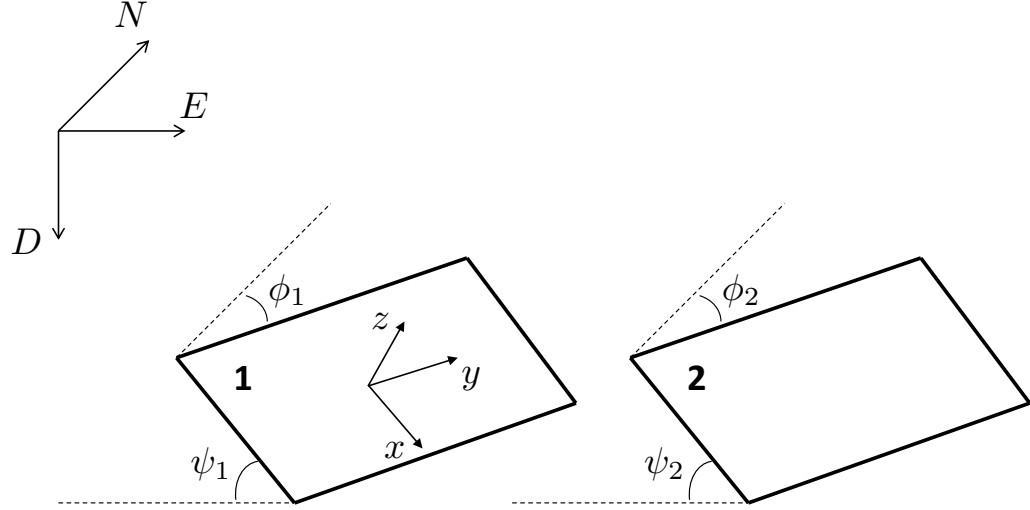





## Tables

**Table 1.** Final slip amplitudes $U_1$ and $U_2$ of asperity 1 and 2 and seismic moment $M_0$ during a seismic event made up of $n$ slipping modes, as a function of the state $P_1$ where the event started. The entry *e.n.* is the abbreviation for *evaluated numerically*.

| State $P_1$ | $n$ | $U_1$ | $U_2$ | $M_0$ |
|---|---|---|---|---|
| $P_1 \in \mathbf{Q_1}$ | 1 | $\kappa U$ | 0 | $\kappa M_1$ |
| $P_1 \in \mathbf{Q_2}$ | 1 | 0 | $\beta\kappa U$ | $\beta\kappa M_1$ |
| $P_1 \in \mathbf{s_1} \vee P_1 \in \mathbf{s_2}$ | 2 | $\kappa U$ | $\beta\kappa U$ | $\kappa M_1(1+\beta)$ |
| $P_1 \in \mathbf{R_1} \vee P_1 \in \mathbf{R_2}$ | 3 | *e.n.* | *e.n.* | *e.n.* |

**Table 2.** Changes in the final slip amplitudes $U_1$ and $U_2$ of asperity 1 and 2 and in the seismic moment $M_0$ associated with the different seismic events predicted by the model, after a stress perturbation from neighbouring faults. The entry *e.n.* is the abbreviation for *evaluated numerically*.

| Kind of event | $\Delta U_1$ | $\Delta U_2$ | $\Delta M_0$ |
|---|---|---|---|
| one-mode 10 | $\Delta\beta_1\kappa U$ | - | $\Delta\beta_1\kappa M_1$ |
| one-mode 01 | - | $\Delta\beta_2\kappa U$ | $\Delta\beta_2\kappa M_1$ |
| two-mode 10-01/01-10 | $\Delta\beta_1\kappa U$ | $\Delta\beta_2\kappa U$ | $\kappa M_1(\Delta\beta_1 + \Delta\beta_2)$ |
| involving mode 11 | *e.n.* | *e.n.* | *e.n.* |

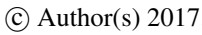

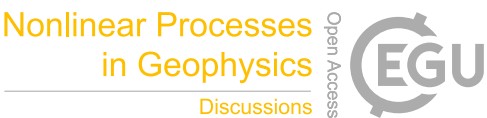

**Table 3.** Future earthquakes generated by the 1992 Landers, California, fault, as functions of the variable $Z_1$ describing the initial state of the 1992 event, with $Z_1 \in [\tilde{Z}_a, \tilde{Z}_b] = [-1.16, 2.19]$. The results predicted by the model before and after the stress perturbation associated with the 1999 Hector Mine, California, earthquake are shown. The values $Z_1 = Z_c = 0.78$ and $Z_1 = Z_c' = 0.50$ correspond to the largest possible earthquakes before and after the stress perturbation, respectively.

| Future earthquake | Unperturbed condition | Perturbed condition |
|---|---|---|
| 1-mode event 01 | $\tilde{Z}_a \leq Z_1 < 0.71$ | $\tilde{Z}_a \leq Z_1 < 0.43$ |
| 2-mode event 01-10 | $Z_1 = 0.71$ | $Z_1 = 0.43$ |
| 3-mode event 01-11-01 | $0.71 < Z_1 < Z_c$ | $0.43 < Z_1 < Z_c'$ |
| 2-mode event 11-01 | $Z_1 = Z_c$ | $Z_1 = Z_c'$ |
| 3-mode event 10-11-01 | $Z_c < Z_1 < 0.92$ | $Z_c' < Z_1 < 0.64$ |
| 2-mode event 10-01 | $Z_1 = 0.92$ | $Z_1 = 0.64$ |
| 1-mode event 10 | $0.92 < Z_1 \leq \tilde{Z}_b$ | $0.64 < Z_1 \leq \tilde{Z}_b$ |





## Appendix A:  Estimate of the stress perturbation

We consider two plane faults, namely fault 1 and fault 2, embedded in an infinite, homogeneous and isotropic Poisson medium of rigidity $\mu$ (Fig. A1). Following the slip of fault 1 (perturbing fault), stress is transferred to fault 2 (receiving fault). We calculate the normal traction $\sigma_n$ and the tangential traction in the direction of slip $\sigma_t$ transferred to the receiving fault, estimated as the average value at its centre.

We define a coordinate system $(x, y, z)$ with axes corresponding with the directions of dip, strike and normal on fault 1, respectively. Fault 1 lies on the plane $z = 0$ and its centre is in the origin of the coordinate system. Accordingly, the unit vector perpendicular to fault 1 is $n_{1i} = (0, 0, 1)$. We call $\phi_1$, $\psi_1$ and $\lambda_1$ the strike, dip and rake angles of fault 1, respectively. The slip direction of fault 1 is then given by

$$m_{1i} = (-\sin \lambda_1, \cos \lambda_1, 0). \tag{A1}$$

Fault 2 is characterized by strike and dip angles $\phi_2$ and $\psi_2$, respectively. Accordingly, the unit vector perpendicular to fault 2 is given by

$$n_{2i} = (\sin \Delta\psi \cos \Delta\phi, -\sin \Delta\psi \sin \Delta\phi, \cos \Delta\psi) \tag{A2}$$

where

$$\Delta\phi = \phi_2 - \phi_1, \qquad \Delta\psi = \psi_2 - \psi_1. \tag{A3}$$

Let $\lambda_2$ be the preferred rake angle on fault 2, correlated with the orientation of tectonic loading: $\lambda_2 = 0°$ for left-lateral strike-slip, $\lambda_2 = 180°$ for right-lateral strike-slip, $\lambda_2 = -90°$ for normal dip-slip and $\lambda_2 = 90°$ for reverse dip-slip. The corresponding slip direction is

$$m_{2x} = \cos \lambda_2 \sin \Delta\phi - \sin \lambda_2 \cos \Delta\psi \cos \Delta\phi \tag{A4}$$

$$m_{2y} = \cos \lambda_2 \cos \Delta\phi + \sin \lambda_2 \cos \Delta\psi \sin \Delta\phi \tag{A5}$$

$$m_{2z} = \sin \lambda_2 \sin \Delta\psi. \tag{A6}$$

We name $(E_i, N_i)$ and $D_i$ the UTM coordinates and depths of the centres of the faults, respectively. In our reference system, the coordinates of the centre of fault 2 are identified by the following three steps:

1. placing the origin at the centre of fault 1:

$$x' = E_2 - E_1, \qquad y' = N_2 - N_1, \qquad z' = D_2 - D_1 \tag{A7}$$





2. clockwise rotation about the $z$ axis by the angle $\phi_1$:

$$x'' = x' \cos\phi_1 - y' \sin\phi_1 \qquad y'' = x' \sin\phi_1 + y' \cos\phi_1, \qquad z'' = z' \tag{A8}$$

3. counterclockwise rotation about the $y$ axis by the angle $\psi_1$:

$$x = x'' \cos\psi_1 - z'' \sin\psi_1, \qquad y = y'', \qquad z = x'' \sin\psi_1 + z'' \cos\psi_1. \tag{A9}$$

The perturbing fault is treated as a point-like dislocation source (a double-couple of forces) located at the origin. This is good approximation for nonoverlapping regions (Dragoni and Lorenzano, 2016). Let $m_0$ be the scalar seismic moment of the dislocation. The $i$-th component of the static displacement field generated by the slip of fault 1 is

$$u_i = -M_{jk} G_{ij,k} \tag{A10}$$

where $M_{ij}$ is the moment tensor associated with the dislocation source

$$M_{ij} = m_0 \left( m_{1i} n_{1j} + m_{1j} n_{1i} \right) \tag{A11}$$

and $G_{ij}$ is the Somigliana tensor

$$G_{ij} = \frac{1}{8\pi\mu} \left( \frac{2}{r} \delta_{ij} - \frac{2}{3} r_{,ij} \right) \tag{A12}$$

with

$$r = \sqrt{x^2 + y^2 + z^2}. \tag{A13}$$

The components of the stress field are given by

$$\sigma_{ij} = \mu(e_{kk}\delta_{ij} + 2e_{ij}), \tag{A14}$$

where $e_{ij}$ is the strain field associated with the displacement field (A10). Finally, the normal traction $\sigma_n$ and the tangential traction in the direction of slip $\sigma_t$ on fault 2 are

$$\sigma_n = \sigma_{ij} n_{2i} n_{2j}, \qquad \sigma_t = \sigma_{ij} m_{2i} n_{2j}. \tag{A15}$$

The signs of $\sigma_n$ and $\sigma_t$ define the effect of the stress transfer on fault 2. If $\sigma_n > 0$, the amount of compressional stress on the receiving fault is reduced, and vice-versa. If $\sigma_t > 0$, the slip of the receiving fault is promoted, and vice-versa.