# Peer review of "Complex interplay between stress perturbations and viscoelastic relaxation in a two-asperity fault model"

_Nonlinear Processes in Geophysics, 2017_

## Short Comment (SC1) · 3 Jan 2018

The manuscript "Complex interplay between stress perturbations and viscoelastic relaxation in a two-asperity fault model" by Lorenzano and Dragoni presents an analysis of two slip asperities separated by a viscoelastic medium using a discrete dynamic system. The model describes the interactions between the two asperities considering the time-dependent response of the medium. The analysis is clear, although the details of the model are beyond my expertise. Figures 2-8 need axis labels in English to improve clarity. I have reviewed two previous versions of the manuscript that were submitted to other journals. The authors have incorporated my previous comments. I suggest
publication after minor revision.

---

## Author Comment (AC1) · 8 Jan 2018

*In the following, figure and page numbers refer to the Interactive Discussion version of the manuscript.*

We are grateful to Sylvain Barbot for his favourable comments on the first version of the manuscript. As for his suggestions regarding Figures 2 – 8, we would like to point out that an English description of the content of each figure is provided in the corresponding caption, which is presented at page 22. Nevertheless, in order to improve clarity, all captions shall be revised and further explanations of the symbols appearing in the

figures shall be provided in the revised version of the manuscript.

---

## Referee Comment (RC1) · Anonymous Referee #1 · 9 Jan 2018

This manuscript describes an effort of using a theoretical asperity model (mostly mathematical without much physical mechanism being invoked) to investigate the effects of viscoelastic relaxation on the evolution of the states of a two-asperity fault in response to the stress perturbation induced by another earthquake. The major conclusion is that earth rheology plays an important role in affecting various features of the two-asperity fault and its future rupture, such as the interseismic interval, the amount of slip (seismic moment), the sequence of the failures of the two asperities, etc. The authors then use the 1992 Landers and 1999 Hector Mine earthquakes as an example to study how the stress perturbation induced by the 1999 earthquake and the viscoelastic deformation associated with the 1992 earthquake interact in determining the characteristics of the

future rupture of the Landers fault. While the scientific question discussed by this paper is very interesting and this work does provide some valuable insights, I think this manuscript can be improved in the following two major aspects.

First, the theoretical model used in this work is based on some assumptions that may not establish in the actual world. Based on my understanding, two major assumptions are listed as follows. While I understand that it is crucial to make these assumptions for the subsequent reasoning, I think the authors should at least discuss how the potential violation of these assumptions would affect the model results.

1. This model appears to assume relatively uniform or similar ruptures of the same asperity throughout multiple earthquake cycles. In the actual earth, earthquake cycles can be irregular, i.e. consecutive events may have different degrees of stress drop. Therefore, the "dynamic friction" may end up at quite different values in different events.

2. This model seems to assume that the asperity part of the fault only shows two modes: sticking and slipping seismically, corresponding to the static and dynamic frictions, respectively; and the part of the fault outside of the asperity only shows steady-state creep (at a constant rate). How would episodic slow slip (faster than relative plate motion) affect the model results? What happens if the fault asperity can slip aseismically?

Second, as an observation-based modeling geophysicist, I found many discussions of this paper to be difficult to understand. This is due to two reasons: (1) the details of the model are actually beyond my expertise; (2) the author may not have done an adequate job, either in the Introduction or at the beginning of Section 2, in linking the mathematical or geometrical terms massively used in the paper with geophysical concepts. For example,

Line 4-5 of Page 2: "dynamics of the system", " means of orbits in the phase space", "dynamic modes"

Line 18-19 of Page 2: "the state of the system at the beginning of the event" To have a broader impact, it is important to define these terms when they are brought up. I also wonder if it's possible to avoid such perplexing terms in the abstract, such as "a vector in the state space" and "state variables of the system", but try to provide a more geophysically meaningful summary instead.

Also, the meanings of the following statements are not clear to me. For example,

Line 3 of Page 1: "variation of their difference". Is that temporal or spatial variation?

Line 14 of Page 2: "the impact of viscoelastic relaxation has first been studied by Amendola and Dragoni (2013) . . .". Do the authors mean "has first been studied in this kind of asperity model"?

Line 30-31 of Page 2: "viscoelastic relaxation on the fault was dealt with by adding a third state variable, the variation in the difference between the slip deficits of the asperities during interseismic intervals." I do not understand the physical mechanism of the process discussed here, any elaboration on that? What does "viscoelastic relaxation on the fault" mean? Are we still talking about mantle viscoelasticity, or ductile deformation of the fault zone? How would creeping behavior (slow slip) of the fault affect this third state variable?

Line 7-8 of Page 4: "the terms $\pm\alpha Z$ are the contribution of stress transfer between the asperities, in the presence of viscoelastic relaxation. . . The parameter $\alpha$ is a measure of the degree of coupling between the asperities." I do not fully understand the physical meaning of this $\alpha$ here. Are the authors suggesting treating partial coupling (creep) of the fault also as a type of viscoelastic behavior?

Line 26 of Page 6: "In the particular case in which P1 belongs to the edge CD, the earthquake will be a two-mode event 11-01." If I understand correctly, this sentence is saying that for an earthquake in which two asperities start to fail at the same time, the weaker asperity would have a longer rupture duration. What are the physical reasons

for that?

Line 16-20 of Page 18: the authors mentioned the effects of stress perturbation on the interseismic intervals of asperities 1 and 2. However, in Line 17-19 of Page 19 (Conclusions), the authors concluded that "the presence of viscoelastic relaxation prevents any prediction about the change in the interseismic time of this fault..." Maybe I didn't understand these, but I found the two discussions contradictory.

Figure 4 is difficult to understand. Can it be clarified?

---

## Author Comment (AC2) · 15 Jan 2018

*We answer point-by-point to the reviewer's comments and suggestions. In the following, equation, figure, page and section numbers refer to the Interactive Discussion version of the manuscript.*

On the whole, the reviewer finds it difficult to relate the mathematical and geometrical terms used in the manuscript with their geophysical meaning. In order to make the text clearer, we shall introduce further explanations in the abstract, introduction and section 2.

[Figure]

Specific comments:

1) This model appears to assume relatively uniform or similar ruptures of the same asperity throughout multiple earthquake cycles. In the actual earth, earthquake cycles can be irregular, i.e. consecutive events may have different degrees of stress drop. Therefore, the "dynamic friction" may end up at quite different values in different events.

The reviewer suggests that, after a seismic event, the values of friction on the fault might be different from the initial ones. This is a possibility, even though it is probable that the change is remarkable only after several seismic cycles. We neglect this possible change, because we focus on other sources of irregularity in the seismic cycles. In fact, seismic cycles are already irregular in the model, for the following reasons: a) heterogeneity of the initial stress on the fault; b) viscoelastic relaxation; c) stress perturbations due to slip on other faults. Consequently, each event is characterized by a different sequence of dynamic modes, with a different stress drop, and the durations of interseismic intervals are variable. However, the model could easily incorporate a change in friction after each event: new values could be given to static and dynamic frictions after the event and the subsequent evolution could be calculated accordingly. But, to our knowledge, there are no available data on changes in friction for real events (apart from laboratory experiments that can be hardly transferred to geologic faults).

2) This model seems to assume that the asperity part of the fault only shows two modes: sticking and slipping seismically, corresponding to the static and dynamic frictions, respectively; and the part of the fault outside of the asperity only shows steady- state creep (at a constant rate). How would episodic slow slip (faster than relative plate motion) affect the model results? What happens if the fault asperity can

slip aseismically?

The reviewer suggests that some regions of the fault may slip aseismically and this may affect the evolution of the fault. This is certainly true. This aspect has been treated in the framework of a discrete fault model by Dragoni and Lorenzano (2017), who considered a region slipping aseismically for a finite time interval and calculated the effect on the stress distribution and the subsequent evolution of the fault. Of course, if the amplitude of aseismic slip has the same order of magnitude as that of seismic slip, the fault evolution is sensibly affected. But, as mentioned above, in the present model we decided to study the role played by other mechanisms.

3) Line 3 of Page 1: "variation of their difference". Is that temporal or spatial variation?

The state variable $Z$ represents the temporal variation in the difference $Y - X$ between the slip deficits of the asperities during interseismic intervals of the fault, due to the stress redistribution associated with viscoelastic relaxation in the asthenosphere. Assuming that the asthenosphere behaves like a Maxwell body of characteristic time $\Theta$, the evolution of $Z$ over time $T$ during an interseismic interval, corresponding to mode 00, is expressed by (Dragoni and Lorenzano, 2015)

$$Z = \bar{Z}e^{-T/\Theta} \tag{1}$$

where $\bar{Z}$ is the value at the beginning of the interseismic interval.

4) Line 14 of Page 2: "the impact of viscoelastic relaxation has first been studied by Amendola and Dragoni (2013)..". Do the authors mean "has first been studied in this kind of asperity model"?

[Figure]

Indeed, Amendola and Dragoni (2013) and Dragoni and Lorenzano (2015) studied viscoelastic relaxation in the framework of a discrete fault model, on which the present work is based upon.

5) Line 30-31 of Page 2: "viscoelastic relaxation on the fault was dealt with by adding a third state variable, the variation in the difference between the slip deficits of the asperities during interseismic intervals." I do not understand the physical mechanism of the process discussed here, any elaboration on that? What does "viscoelastic relaxation on the fault" mean? Are we still talking about mantle viscoelasticity, or ductile deformation of the fault zone? How would creeping behavior (slow slip) of the fault affect this third state variable?

As explained in replying to comment 3, the post-seismic mechanism considered in the present model is the viscoelastic relaxation in the asthenosphere. As for its interaction with slow slip events on the fault, the relaxation definitely transfers stress to the various regions of the fault and may therefore trigger aseismic slip of creeping zones. However, the stress redistribution associated with viscoelastic relaxation becomes significant over times much longer than the typical duration of slow slip events, so that its effect can reasonably be neglected. On the other hand, further research is required to discuss the interaction between viscoelastic relaxation and stable creep in the framework of a discrete fault model; this kind of analysis is beyond the scope of the present work, but it may be object of future research by combining elements of the present model with the model of Dragoni and Lorenzano (2017).

6) Line 7-8 of Page 4: "the terms $\pm\alpha Z$ are the contribution of stress transfer between the asperities, in the presence of viscoelastic relaxation...The parameter $\alpha$ is a measure of the degree of coupling between the asperities." I do not fully understand the physical meaning of this $\alpha$ here. Are the authors suggesting treating partial coupling

(creep) of the fault also as a type of viscoelastic behavior?

According to Eq. (2), the tangential force acting on each asperity in the slip direction is made up of two terms. The first term is related with the effect of tectonic loading, taking place at constant rate; the second term, where the parameter $\alpha$ becomes involved, represents the stress transfer between the asperities. In the framework of the present model, stress is transferred by one asperity to the other as a result of coseismic slip, corresponding to any one of the dynamic modes 10, 01 and 11; in the subsequent interseismic interval (mode 00), the static stress field generated by asperity slip undergoes a certain amount of relaxation owing to viscoelasticity (see reply to comment 3). The parameter $\alpha$ conveys the strength of coupling between the asperities: for smaller values of $\alpha$, the stress transfer from one asperity to the other is less efficient. In the limit case $\alpha = 0$, the asperities are completely independent from one another and the slip of one of them does not affect the state of the other: the evolution of the asperities is thus governed by tectonic loading only. By comparison with a model based on continuum mechanics, the specific value of $\alpha$ can be estimated as (Dragoni and Tallarico, 2016)

$$\alpha = \frac{Avs}{2\dot{e}} \tag{2}$$

where $A$ is the area of the asperities, $v$ is the velocity of the tectonic plates, $s$ is the tangential traction (per unit moment) imposed on one asperity by the slip of the other and $\dot{e}$ is the tangential strain rate on the fault due to tectonic loading.

7) Line 26 of Page 6: "In the particular case in which P1 belongs to the edge CD, the earthquake will be a two-mode event 11-01." If I understand correctly, this sentence is saying that for an earthquake in which two asperities start to fail at the same time, the weaker asperity would have a longer rupture duration. What are the physical reasons for that?

By definition, asperity 2 is weaker than asperity 1; that is, friction on asperity 2 is smaller than friction on asperity 1 ($0 < \beta < 1$). If the asperities start slipping simultaneously (so that the system passes from mode 00 to mode 11), asperity 1 is bound to stop the first, while asperity 2 continues to slip. As a result, mode 11 is followed by mode 01 and the slip of the weaker asperity has a longer duration. The opposite would hold if asperity 2 were stronger than asperity 1 ($\beta > 1$), so that the slip event resulting from initial states belonging to the edge $CD$ of the sticking region would be a two-mode event 11 - 10.

8) Line 16-20 of Page 18: the authors mentioned the effects of stress perturbation on the interseismic intervals of asperities 1 and 2. However, in Line 17-19 of Page 19 (Conclusions), the authors concluded that "the presence of viscoelastic relaxation prevents any prediction about the change in the interseismic time of this fault..." Maybe I didn't understand these, but I found the two discussions contradictory.

Following a stress perturbation due to earthquakes on neighbouring faults, an increase in the Coulomb stress associated with a given asperity directly yields the anticipation of the slip of that asperity, and vice-versa, if a purely elastic rheology is assumed for the receiving fault (section 4.1.3). According to the present model, this property no longer holds if the change in Coulomb stress occurs while viscoelastic relaxation is taking place on the receiving fault. In fact, even if the change in the interseismic intervals of the asperities can still be evaluated from a theoretical point of view (sections 4.1.2 and 5.3), the specific effect of the stress perturbation could be univocally inferred only if the particular states of the fault at the time of the stress perturbation and right after it were known. The information on the change in Coulomb stress on the fault do not suffice any more. Such complication and the consequent unpredictability of the net effect of a stress perturbation is exemplified in section 5, where we show that the consequences

of the 1999 Hector Mine, California, earthquake on the post-seismic evolution of the 1992 Landers, California, fault depend on the specific state of the Landers fault at the time of the 1999 earthquake and immediately after it, even if the variations in the Coulomb stress on the asperities are known.

9) Figure 4 is difficult to understand. Can it be clarified?

In Figure 4, the faces $AECD$ and $BCDF$ of the sticking region of the system, where seismic events start, are shown. As discussed at page 6, lines $20-26$, these faces can be divided into different subsets (trapezoids and segments), each one corresponding to a specific sequence of dynamic modes during the seismic event. In order to improve clarity, we shall expand the caption of the figure and list all seismic events resulting from the various subsets.

**References**

Amendola, A. and Dragoni, M. (2013). Dynamics of a two-fault system with viscoelastic coupling, Nonlinear Process. Geophys., 20(1), $1-10$.

Dragoni, M. and Lorenzano, E. (2015). Stress states and moment rates of a two-asperity fault in the presence of viscoelastic relaxation, Nonlinear Process. Geophys., 22(3), $349-359$.

Dragoni, M. and Lorenzano, E. (2017). Dynamics of a fault model with two mechanically different regions, Earth Planets Space, 69(145), doi:10.1186/s40623-017-0731-2.

Dragoni, M. and Tallarico, A. (2016). Complex events in a fault model with interacting asperities, Phys. Earth Planet. Inter., 257, $115-127$.